# Deformation-aware GAN for Medical Image Synthesis with Substantially Misaligned Pairs

**Bowen Xin**[1]                                                     bowen.xin@csiro.au
**Tony Young**[2]                                          Tony.Young@health.nsw.gov.au
**Claire E Wainwright**[3,4]                      Claire.Wainwright@health.qld.gov.au
**Tamara Blake**[3,4]                              Tamara.Blake@health.qld.gov.au
**Leo Lebrat**[5]                                             leo.lebrat@csiro.au
**Thomas Gaass**[6]                                        thomas.gaass@gmail.com
**Thomas Benkert**[7]                      benkert.thomas@siemens-healthineers.com
**Alto Stemmer**[7]                          alto.stemmer@siemens-healthineers.com
**David Coman**[4]                               David.Coman@health.qld.gov.au
**Jason Dowling**[1]                                       jason.dowling@csiro.au

[1] *Australian e-Health Research Centre, CSIRO, Brisbane, Australia*

[2] *Liverpool and Macarthur Cancer Therapy Centres and Ingham Institute, Sydney, Australia*

[3] *Child Health Research Centre, University of Queensland, Brisbane, Australia*

[4] *Queensland Children's Hospital, Brisbane, Australia*

[5] *Data61, CSIRO, Brisbane, Australia*

[6] *Siemens Healthcare Pty Ltd, Brisbane, Australia*

[7] *Siemens Healthineers AG, Erlangen, Germany*

**Editors:** Accepted for publication at MIDL 2024

## Abstract

Medical image synthesis generates additional imaging modalities that are costly, invasive or harmful to acquire, which helps to facilitate the clinical workflow. When training pairs are substantially misaligned (e.g., lung MRI-CT pairs with respiratory motion), accurate image synthesis remains a critical challenge. Recent works explored the directional registration module to adjust misalignment in generative adversarial networks (GANs); however, substantial misalignment will lead to 1) suboptimal data mapping caused by correspondence ambiguity, and 2) degraded image fidelity caused by morphology influence on discriminators. To address the challenges, we propose a novel Deformation-aware GAN (DA-GAN) to dynamically correct the misalignment during the image synthesis based on multi-objective inverse consistency. Specifically, in the generative process, three levels of inverse consistency cohesively optimise symmetric registration and image generation for improved correspondence. In the adversarial process, to further improve image fidelity under misalignment, we design deformation-aware discriminators to disentangle the mismatched spatial morphology from the judgement of image fidelity. Experimental results show that DA-GAN achieved superior performance on a public dataset with simulated misalignments and a real-world lung MRI-CT dataset with respiratory motion misalignment. The results indicate the potential for a wide range of medical image synthesis tasks such as radiotherapy planning.

## 1. Introduction

Medical image synthesis produces additional imaging modalities to provide essential information for diagnosis or treatment planning, while bypassing the cost and extra time associated with additional scans. It is particularly useful when the additional scan is invasive, harmful, costly or time-consuming (Liu et al., 2022). Typical applications include synthetic CT for MRI-only radiotherapy dose planning (Spadea et al., 2021) or children's airway assessment (Longuefosse et al., 2023)). Generative adversarial networks (GANs) are widely used in medical synthesis, which usually requires either well-aligned imaging pairs (by supervised methods) or randomly unpaired data (by unsupervised methods). Specifically, supervised GANs, such as Pix2pix and its improved variants (Wang et al., 2018a,b; AlBahar and Huang, 2019), leverage pixel-wise loss on well-aligned imaging pairs to learn the unique and optimal mapping. However, well-aligned pairs are not widely available due to patient motion or organ movement, causing accumulated error and unreasonable placement in supervised methods (Pang et al., 2021). Though registration is commonly used as preprocessing to align images, it is still difficult to acquire perfectly aligned pairs, especially under substantial misalignment such as respiratory motion in lung MR-to-CT synthesis (Sotiras et al., 2013).

Unsupervised GANs are not ideal for misaligned pairs either. Specifically, unsupervised GANs enable training on randomly unpaired data by leveraging extra constraints such as cycle consistency (Zhu et al., 2017; Hoffman et al., 2018; Khorram and Fuxin, 2022), mutual information (Park et al., 2020; Zhan et al., 2022), or geometry consistency (Fu et al., 2019; Xu et al., 2022). However, they are not designed to utilise pairing information to uncover optimal mappings (minimised pixel-wise error), which is essential in tasks such as radiotherapy planning. According to (Shen et al., 2020), cycle consistency mapping used in unsupervised GANs is not strictly one-to-one mapping, which is an important condition in intra-subject medical image synthesis (Wang et al., 2021). Diffusion models have shown great potential in computer vision applications due to their strength in capturing distributions (Ho et al., 2020; Song et al., 2020); however, they are computationally expensive and data-hungry to train, hindering their application in the medical domain.

One recent work RegGAN (Kong et al., 2021) explored directional registration in image synthesis on datasets with simulated misalignment; however, the real-life setting often involves large deformation between pairs (e.g., Figure 1), causing difficulty in learning unique one-to-one mapping due to a large number of local minima (Christensen and Johnson, 2001). The resulting correspondence ambiguity and asymmetric mapping error would add to pixel-wise error in supervised methods, causing a major challenge in generative modelling. The second challenge is the degraded image fidelity caused by the influence of spatial misalignment during the adversarial process. To further elaborate on the issue, the discriminator in RegGAN may recognise the real/fake images purely based on spatial morphology rather than intensity characteristics, thus leading to suboptimal image fidelity.

In this paper, we propose a Deformation-aware GAN (DA-GAN) to jointly address the above two synthesis challenges when image pairs are substantially misaligned. Firstly, inspired by the capacity of symmetric registration to jointly estimate invertible bidirectional transformation, we propose a multi-objective inverse consistency to comprehensively investigate how to cohesively incorporate symmetric registration into an image generation network. To further improve degraded image fidelity in an adversarial process, we design a

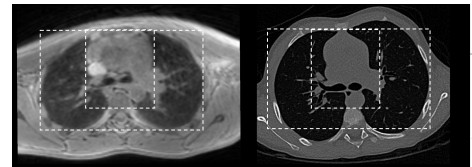

Figure 1: Substantial misalignment in lung MRI-CT pairs due to respiratory motion.

deformation-aware adversarial loss, which leverages the outputs from symmetric registration to guide the discriminator to learn the intensity characteristics (that are disentangled from the mismatched spatial morphology). We comprehensively validated our proposed DA-GAN on two datasets, including a public simulation brain dataset with 6 levels of simulated misalignments, and a real-world lung MRI-CT dataset with challenging respiratory-motion misalignments.

## 2. Methodology

**Problem formulation** Suppose we have a training dataset with misaligned imaging pairs $(x_i, y_i)|_{i=1}^n$, where $x_i \in X$ and $y_i \in Y$ belong to different modalities. $X$ and $Y$ differ in both intensity characteristics and spatial morphology. Additionally, we denote $\hat{y}_i \in \hat{Y}$ as a transformed $y_i$ that spatially corresponds to source imaging $x_i$, but $\hat{y}_i$ is unknown in the real world. In other words, both $x_i$ and $\hat{y}_i$ are aligned in source spatial space, but only differ in intensity characteristics. With misaligned multimodal imaging pairs $(x_i, y_i)|_{i=1}^n$, our objective is to accurately synthesise the target imaging $\hat{y}_i$ that is spatially corresponding to the source image $x_i$ for subsequent tasks such as radiotherapy treatment planning.

### 2.1. DA-GAN overview

**DA-GAN network architecture** Figure 2a presents the network architecture of our proposed DA-GAN which consists of three major components, including (1) modality generators $G$ and $F$, (2) symmetric spatial aligners $A_y$ and $A_x$, and (3) deformation-aware discriminators $D_y$ and $D_x$. Firstly, modality generators are designed to translate the source image to the target appearance with spatial correspondence preserved, which is implemented with trainable networks G: $x \to \hat{y} \in \hat{Y}$ and F: $y \to \hat{x} \in \hat{X}$. Secondly, symmetric spatial aligners $A_y$ and $A_x$ are designed to exploit symmetric correspondence during image-to-image translation to optimise unique and optimal mapping. Each aligner (e.g., $A_y = \{R_y^{\rightarrow}, R_y^{\leftarrow}, T\}$) is enforced to learn bidirectional transformations $\phi_y^{\rightarrow} : \hat{y} \to y$ and $\phi_y^{\leftarrow} : y \to \hat{y}$ that are inverse to each other. The bidirectional transformations are learnt through symmetric transformation repressors $R = \{R^{\rightarrow}, R^{\leftarrow}\}$. Each transformation regressor is a CNN model trained to predict a deformation field (Kong et al., 2021), and then followed by a spatial transformer network (Jaderberg et al., 2015) to resample images to target spatial space. Thirdly, deformation-aware discriminators are denoted as $D_y : y \to \{0, 1\}$ where $y \in U(Y, \hat{Y})$ and $D_x : x \to \{0, 1\}$ where $x \in U(X, \hat{X})$.

**DA-GAN objective** To synthesize with misaligned pairs, DA-GAN is constrained by three loss functions, including (1) symmetric registration loss $L_{sr}$ (in Figure 2a) for self-aligning, (2)

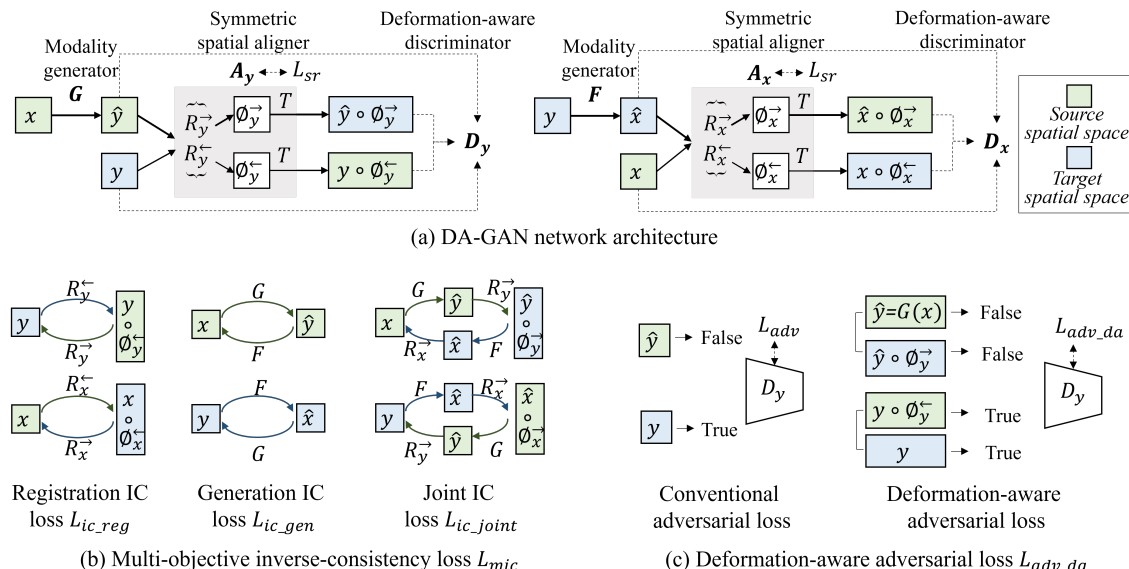

(a) DA-GAN network architecture

(b) Multi-objective inverse-consistency loss $L_{mic}$

(c) Deformation-aware adversarial loss $L_{adv\_da}$

Figure 2: (a) Network architecture of DA-GAN. (b) $L_{mic}$ loss dynamically enhances image correspondence from three objectives. (c) $L_{adv\_da}$ loss guides discriminators to learn deformation for improved image fidelity.

multi-objective inverse-consistency loss $L_{mic}$ (Figure 2b) for aligning enhancement, and (3) deformation-aware adversarial loss $L_{adv\_da}$ (Figure 2c) for synthesis enhancement. Overall, we can formulate our total loss function for DA-GAN as below:

$$\min_{G,F,R} \max_{D} L_{total}(G, F, R, D) = L_{sr} + L_{mic} + L_{adv\_da} \tag{1}$$

where $R = \{R_y, R_x\}$ and $D = \{D_y, D_x\}$. These three loss functions will be introduced in the following sections, respectively.

## 2.2. DA-GAN loss functions

**Symmetric registration loss** $L_{sr}$ is designed to (1) punish dissimilarity between misaligned imaging pairs and (2) encourage local smoothness on the deformation field. The former similarity loss $L_{sim}$ for symmetric registration is defined as:

$$\min_{G,F,R_y,R_x} L_{sim}(G, F, R_y, R_x) = \mathbb{E}_{x,y}[||y - G(x) \circ \phi_y^{\rightarrow}||_1 + ||G(x) - y \circ \phi_y^{\leftarrow}||_1$$
$$+ ||x - F(y) \circ \phi_x^{\rightarrow}||_1 + ||F(y) - x \circ \phi_x^{\leftarrow}||_1] \tag{2}$$

where $\phi_y^{\rightarrow} = R_y^{\rightarrow}(G(x), y)$, $\phi_y^{\leftarrow} = R_y^{\leftarrow}(y, G(x))$, $\phi_x^{\rightarrow} = R_x^{\rightarrow}(F(y), x)$, $\phi_y^{\leftarrow} = R_y^{\leftarrow}(x, F(y))$. Secondly, the smoothness loss $L_{smt}$ (Balakrishnan et al., 2019) is implemented to minimize the gradient divergence of the estimated deformation field:

$$\min_{R_y,R_x} L_{smt}(R_y, R_x)) = \mathbb{E}_{x,y}[||\nabla \phi_y^{\rightarrow}||^2 + ||\nabla \phi_y^{\leftarrow}||^2 + ||\nabla \phi_x^{\rightarrow}||^2 + ||\nabla \phi_x^{\leftarrow}||^2] \tag{3}$$

Lastly, by integrating Equation 2 and 3 with loss weights $\lambda_{reg}$ and $\lambda_{smt}$, we can formulate our symmetric registration loss $L_{sr}$ as below:

$$L_{sr} = \lambda_{reg}L_{reg} + \lambda_{smt}L_{smt} \tag{4}$$

**Multi-objective inverse-consistency loss** $L_{mic}$ is proposed to (1) improve image alignment during symmetric registration, and (2) improve synthesis correspondence to the source image during image generation. As illustrated in Figure 2b, this is achieved by enforcing inverse consistency from three different levels, including registration level, generation level, and joint level.

Firstly, at the registration level, we enforce inverse consistency on the forward and backward transformations $\phi^{\rightarrow}$ and $\phi^{\leftarrow}$ during symmetric registration. Specifically, the registration IC loss $L_{ic\_reg}$ is formulated as

$$\min_{R_y,R_x} L_{ic\_reg}(R_y, R_x) = E_{x,y}[||y \circ \phi_y^{\leftarrow} \circ \phi_y^{\rightarrow} - y||_1 + ||x \circ \phi_x^{\leftarrow} \circ \phi_x^{\rightarrow} - x||_1] \tag{5}$$

Secondly, at the generation level, we constrain inverse consistency on the two modality generators G and F. Thus, the generation IC loss $L_{ic\_gen}$ is formulated as below:

$$\min_{G,F} L_{ic\_gen}(G, F) = \mathbb{E}_x[||F(G(x)) - x||_1 + \mathbb{E}_y[||G(F(y)) - y||_1] \tag{6}$$

Lastly, we propose a third joint level inverse consistency through both image registration and generation cycle, thus jointly optimising image registration and generation. The formulation of joint inverse-consistency $L_{ic\_joint}$ is shown as below:

$$\min_{G,F,R_y,R_x} L_{ic\_joint}(G, F, R_y, R_x) = \mathbb{E}_{x,y}[||F(G(x) \circ \phi_y^{\rightarrow}) \circ \phi_x^{\rightarrow} - x||_1$$
$$+ ||G(F(y) \circ \phi_x^{\rightarrow}) \circ \phi_y^{\rightarrow} - y||_1] \tag{7}$$

To summarise, the overall multi-objective inverse-consistency loss is composed of three levels of inverse consistency (with their corresponding weights denoted as $\lambda$):

$$L_{mic} = \lambda_{ic\_reg}L_{ic\_reg} + \lambda_{ic\_gen}L_{ic\_gen} + \lambda_{ic\_joint}L_{ic\_joint} \tag{8}$$

**Deformation-aware adversarial loss** $L_{adv\_da}$ is designed to disentangle the influence of spatial morphology across domains from intensity characteristic learning. We illustrate the comparison of conventional adversarial loss $L_{adv}$ and our $L_{adv\_da}$ for an example discriminator $D_y$ in Figure 2c. Via symmetric spatial aligner $A_y$, we can obtain source-shaped images and target-shaped images for both generated ($G(x)$ and $G(x) \circ \phi_y^{\rightarrow}$) and real images ($y \circ \phi_y^{\leftarrow}$ and $y$). Then, we feed all these images to the discriminator in the adversarial process to guide it to focus on learning intensity characteristics only. Formally, deformation-aware adversarial loss for $D_y$ is formulated as:

$$\min_G \max_{D_y} L_{adv\_da}^y(G, D_y) = \mathbb{E}_y[log(D_y(y)) + log(D_y(y \circ \phi_y^{\leftarrow})))]$$
$$+ \mathbb{E}_x[log(1 - D_y(G(x))) + log(1 - D_y(G(x) \circ \phi_y^{\rightarrow}))] \tag{9}$$

Similarly, we can derive the other half of adversarial loss for $D_x$ as $L_{adv\_da}^x$. The total adversarial loss $L_{adv\_da} = \lambda_{adv\_da}L_{adv\_da}^y + \lambda_{adv}L_{adv\_da}^x$ where $\lambda_{adv\_da}$ denotes loss weight.

## 3. Experiments

**Simulation dataset**   For simulation experiments, the public brain T1-T2 MRI dataset (BraTS 2018 (Menze et al., 2014)) was selected because there were well-aligned imaging pairs as ground truth. The training and testing sets contained 5760 and 768 pairs of T1 and T2 images, respectively. The data were normalised to [-1, 1], resized to 256*256, and publicly available [1] (Kong et al., 2021). As original brain images were paired and well-aligned, we simulated 6 different levels of non-affine misalignments on the dataset.

**Clinical lung MRI/CT dataset**   The private lung MRI-CT dataset was used to validate the proposed algorithm for a real-world radiotherapy treatment planning setting. The dataset contained 4096 pairs of ultrashort-echo time MRI (from Siemens scanners) and CT imaging for training and 1024 pairs for independent testing. Both imaging modalities were normalised to [-1, 1], cropped to lung regions, resampled to isotropic, and preliminarily registered. However, we still observed alignment errors in lung regions (Dice 0.949) as well as bones and airways. Please refer to Appendix A for additional details on both datasets.

**Experiment settings**   In simulation experiments on the brain dataset, 6 non-affine deformations were randomly applied on the training sets to simulate the misalignment. The non-affine deformation was implemented with elastic deformation on control points (Rand2DElastic in MONAI library (Cardoso et al., 2022)) with 6 incremental levels denoted as NA-1 to NA-6. In real-world experiments on the lung dataset, DA-GAN was compared on 8 state-of-the-art (SOTA) medical synthesis methods, including GAN (Goodfellow et al., 2020), Pix2pix (Isola et al., 2017), CycleGAN (Zhu et al., 2017), UNIT (Liu et al., 2017), MUNIT (Huang et al., 2018), NiceGAN (Chen et al., 2020), RegGAN-NC (Kong et al., 2021), and RegGAN-C (Kong et al., 2021). Please find more details in Appendix B.1.

All experiments were implemented in Pytorch on a 64-bit Ubuntu Linux system with one 16 GB Nvidia P100 GPU. We trained all the methods using the Adam optimiser with the learning rate 1e-4 and $(\beta_1, \beta_2) = (0.5, 0.999)$. The batch size was set to 1 with weight decay 1e04. The training included 50 epochs for both datasets. The brain imaging was evaluated with three metrics in 2D, including Normalized Mean Absolute Error (NMAE), Peak Signal-to-Noise Ratio (PSNR) and Structural Similarity (SSIM). The lung imaging was evaluated with MAE3D, PSNR3D, and SSIM3D. The background of the image was excluded from the computation. For reproducibility, we included more implementation details on DA-GAN modules and loss weighting in Appendix B.2.

## 4. Results

**Results on the simulation experiments**   This section summarises the results of DA-GAN on brain imaging with 6 non-affine misalignments (Table 1) and visualisation on the testing set (Figure 3). Table 1 shows that DA-GAN consistently outperformed all comparison methods in three metrics from misalignment levels NA-1 to NA-6. The results of DA-GAN remained stable with increased levels of non-affine misalignment with NMAE ranging from 0.070 to 0.075. Figure 3 visualises that our DA-GAN achieved less error compared with other methods on the error map.

---

1. https://drive.google.com/file/d/1PiTzGQEVV7NO4nPaHeQv61WgDxoD76nL/view

Table 1: Results on the brain dataset with 6 simulated non-affine misalignments.

| Non-affine (NA) | NA-1 | | | NA-2 | | | NA-3 | | |
|---|---|---|---|---|---|---|---|---|---|
| Methods | NMAE | PSNR | SSIM | NMAE | PSNR | SSIM | NMAE | PSNR | SSIM |
| GAN | 0.111 | 20.12 | 0.784 | 0.102 | 21.73 | 0.824 | 0.162 | 16.65 | 0.700 |
| Pix2pix | 0.091 | 21.71 | 0.807 | 0.101 | 17.88 | 0.776 | 0.094 | 20.76 | 0.808 |
| CycleGAN | 0.087 | 23.35 | 0.825 | 0.091 | 23.41 | 0.817 | 0.093 | 22.75 | 0.836 |
| RegGAN-NC | 0.074 | 24.84 | 0.854 | 0.073 | 25.13 | 0.852 | 0.076 | 24.82 | 0.847 |
| RegGAN-C | 0.078 | 24.24 | 0.850 | 0.079 | 24.30 | 0.850 | 0.076 | 23.22 | 0.853 |
| **DA-GAN** | **0.074** | **24.97** | **0.859** | **0.070** | **25.25** | **0.861** | **0.073** | **24.89** | **0.857** |
| Non-affine (NA) | NA-4 | | | NA-5 | | | NA-6 | | |
| Methods | NMAE | PSNR | SSIM | NMAE | PSNR | SSIM | NMAE | PSNR | SSIM |
| GAN | 0.094 | 20.79 | 0.815 | 0.106 | 21.95 | 0.813 | 0.100 | 21.54 | 0.813 |
| Pix2pix | 0.120 | 17.35 | 0.746 | 0.112 | 18.34 | 0.756 | 0.116 | 15.12 | 0.761 |
| CycleGAN | 0.116 | 22.23 | 0.823 | 0.086 | 23.47 | 0.833 | 0.108 | 22.92 | 0.786 |
| RegGAN-NC | 0.077 | 24.46 | 0.839 | 0.082 | 22.16 | 0.828 | 0.081 | 22.33 | 0.817 |
| RegGAN-C | 0.080 | 23.54 | 0.850 | 0.076 | 23.36 | 0.848 | 0.082 | 22.88 | 0.841 |
| **DA-GAN** | **0.072** | **24.58** | **0.858** | **0.075** | **24.72** | **0.858** | **0.073** | **24.61** | **0.858** |

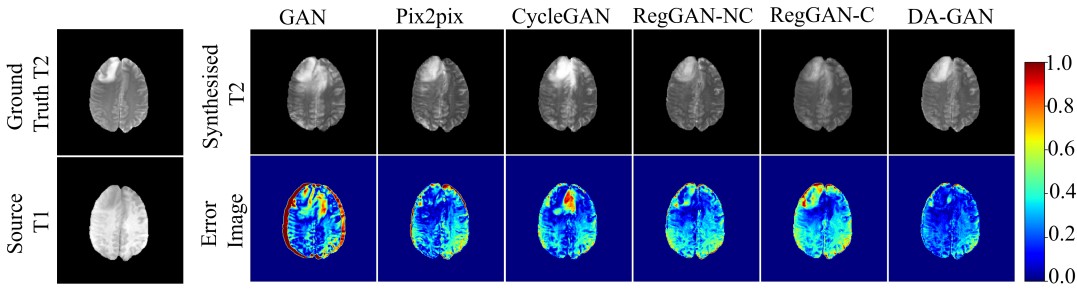

(a) Visualisation of brain dataset with non-affine misalignment NA-3.

Figure 3: Visualisation of prediction and error images in the simulation experiments (NA-3).

**Results on the lung MRI-CT dataset** Table 2 shows the quantitative results of DA-GAN on the lung MRI-CT with challenging respiratory-motion misalignment. The results in Table 2 show that our DA-GAN achieved the best results of MAE3D 35.86, PSNR3D 32.49 and SSIM3D 0.731 compared with 8 SOTA methods. Figure 4 further visually highlights the superiority of our DA-GAN, especially in the challenging spine, bones and heart regions (blue arrows). These demonstrate the great potential for synthetic CT for MRI-only radiotherapy.

**Ablation study on DA-GAN losses** The ablation study in Table 3 shows that $L_{adv\_da}$ improved the baseline RegGAN in all metrics (e.g., PSNR-3D 0.41), while $L_{mic}$ further improved the results in all metrics (e.g., PSNR-3D 0.36). The paired t-test shows that the contributions from both losses were statistically significant (p-value<0.001) in MAE2D, PSNR2D, and SSIM2D. Further details are in Appendix C. Due to space limit, please find the analysis of convergence, complexity and hyperparameter sensitivity in Appendix D-F.

Table 2: Performance comparison on the real-world lung dataset.

| Methods | MAE3D | PSNR3D | SSIM3D |
|---------|-------|--------|--------|
| GAN | $330.66 \pm 21.59$ | $17.00 \pm 0.32$ | $0.612 \pm 0.035$ |
| Pix2pix | $39.64 \pm 8.31$ | $31.88 \pm 1.48$ | $0.710 \pm 0.027$ |
| CycleGAN | $49.32 \pm 4.12$ | $30.14 \pm 0.82$ | $0.691 \pm 0.015$ |
| UNIT | $267.9 \pm 15.15$ | $18.51 \pm 0.32$ | $0.617 \pm 0.034$ |
| MUNIT | $39.78 \pm 6.56$ | $31.62 \pm 1.01$ | $0.705 \pm 0.024$ |
| NiceGAN | $42.73 \pm 8.39$ | $31.21 \pm 1.61$ | $0.711 \pm 0.022$ |
| RegGAN-NC | $39.45 \pm 5.07$ | $31.86 \pm 1.00$ | $0.710 \pm 0.021$ |
| RegGAN-C | $40.77 \pm 4.59$ | $31.72 \pm 0.97$ | $0.710 \pm 0.017$ |
| **DA-GAN** | $\mathbf{35.86 \pm 6.97}$ | $\mathbf{32.49 \pm 1.38}$ | $\mathbf{0.731 \pm 0.023}$ |

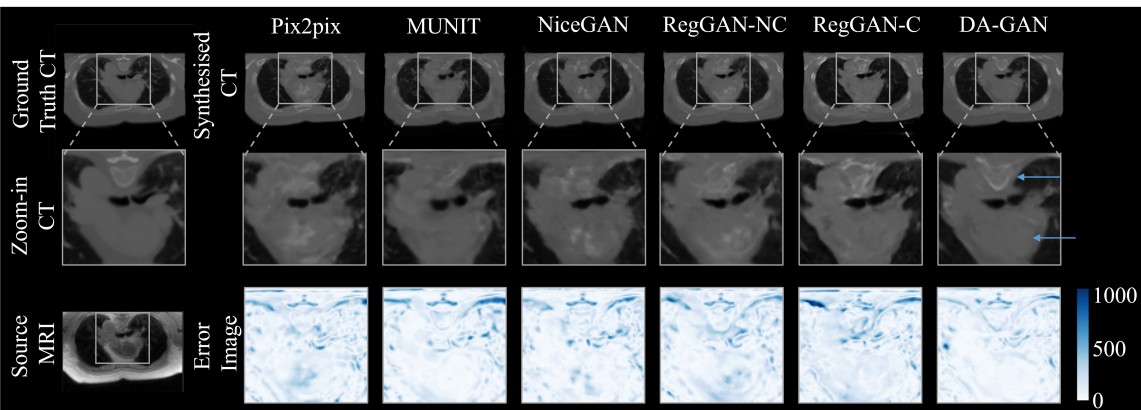

Figure 4: Visualization of synthesised images and error maps on the lung dataset. The blue arrows indicate our DA-GAN achieved better visual results at spine and heart.

Table 3: Ablation study on the proposed losses $L_{adv\_da}$ and $L_{mic}$.

| | MAE2D | PSNR2D | SSIM2D | MAE3D | PSNR3D | SSIM3D |
|---|-------|--------|--------|-------|--------|--------|
| RegGAN-C | 0.011 | 70.23 | 0.918 | 40.77 | 31.72 | 0.710 |
| DA-GAN ($L_{adv\_da}$) | 0.010 | 74.82 | 0.925 | 37.23 | 32.13 | 0.719 |
| DA-GAN ($L_{adv\_da} + L_{mic}$) | **0.009** | **76.36** | **0.929** | **35.86** | **32.49** | **0.725** |

## 5. Conclusions

In this study, we introduce a new DA-GAN for medical image synthesis with substantially misaligned imaging pairs. We propose two novel loss functions $L_{mic}$ and $L_{adv\_da}$ to generate high-fidelity images while adaptively learning correspondence via symmetric registration. We validated our method on a public brain dataset with both 6 simulated misalignments and a real-world lung dataset compared with 8 SOTA methods. The results demonstrated the potential towards an important step in generalisable medical image synthesis with limited data for clinical applications such as early diagnosis and radiotherapy planning.

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

## Appendix A. Dataset details

**Brain T1-T2 dataset** On the brain T1-T2 MRI dataset, we simulated 6 different levels of non-affine misalignments. A visualisation of non-affine misalignment is provided in Figure 5. An overview of implementation parameters for non-affine misalignment is provided in Table 4. More specifically, the non-affine misalignment was simulated using elastic deformation on control points (Rand2DElastic in MONAI library[2]). The spacing between control points was set to [40, 40], while the magnitude was set to incremental levels from NA-1 to NA-6.

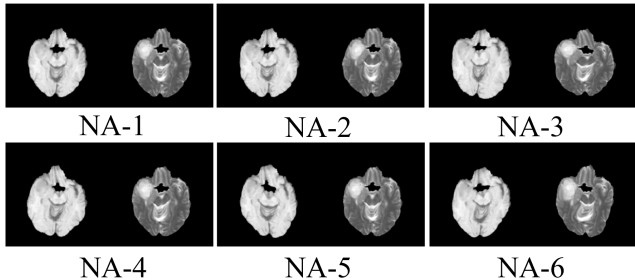

NA-1          NA-2          NA-3

NA-4          NA-5          NA-6

Figure 5: Example images with different levels of non-affine misalignment.

Table 4: Implementation of non-affine misalignments

| Non-affine (NA) | NA-1 | NA-2 | NA-3 | NA-4 | NA-5 | NA-6 |
|---|---|---|---|---|---|---|
| Spacing | [40, 40] | [40, 40] | [40, 40] | [40, 40] | [40, 40] | [40, 40] |
| Magnitude | [1, 2] | [2, 3] | [3,4] | [4, 5] | [5, 6] | [6, 7] |

**Lung MRI-CT dataset** The lung MRI-CT dataset contained paired but misaligned MRI and CT from 20 patients with lung diseases. The lung MRI was implemented with a prototype 3D free-breathing stack-of-spirals UTE VIBE sequence, provided by Siemens Healthineers (Mugler III et al., 2015; Kumar et al., 2017). UTE-MRI provided millimetre resolution and radiation-free assessment for pulmonary structural imaging (Dournes et al., 2016). 20 CTs were scanned on approximately the same day as MRI. 20 patients included 12 patients with cystic fibrosis and 8 patients with lung cancer. For the MRI protocol, both cystic fibrosis and lung cancer cohorts used Siemens 3T scanners with flip angle 5, echo time 0.05, and repetition time 2.97-3.78. For details of image size and spacing, please refer to Table 5. Both MRI and CT were normalised to [-1, 1], cropped to lung regions, resampled to isotropic spacing, and preliminarily registered. The ground-truth paired MRI-CT images were acquired via an automatic registration pipeline including elastic registration (SimpleElastix library), structure-guided registration (demon registration in ants library) and antsRegistrationSyNQuick (ants library). After registration, we still observed registration errors on the whole lung regions with Dice 0.949 (Dice for right lung 0.947, Dice for right lung

---

2. https://docs.monai.io/en/stable/transforms.html#rand2delastic

Table 5: Physical details of the lung MRI-CT dataset

| Lung cancer | Image size | No. slice | Pixel spacing | Slice spacing |
|---|---|---|---|---|
| MRI-spiral vibe | 320*320 | 176-210 | 1.5*1.5 | 1.5 |
| CT | 512*512 | 181-234 | 1.71*1.71 | 2 |
| **Cystic fibrosis** | Image size | No. slice | Pixel spacing | Slice spacing |
| MRI-spiral vibe | 512*512 or 416*416 | 160-241 | 1.1 *1.1 or 1.25*1.25 | 1.1 or 1.25 |
| CT | 512*512 | 267-902 | 0.46*0.46-0.78*0.78 | 0.4-1 |

0.950) and registration errors in bones and airways. We acquired relevant ethics approval for the study, and informed written consent from the parent or legal guardian of each child.

## Appendix B. Implementation details

In this section, we first discuss the implementation of other comparison methods. Then we introduce network implementation details for different modules and loss weighting in DA-GAN.

### B.1. Implementation of comparison methods

**GAN** uses a generator to translate the source image to the target space, while using a discriminator to determine whether images were generated from the real data. In this way, GANs iteratively improve the image fidelity via min-max game. In the implementation, the generator and discriminator shared the same network architecture as ours for a fair comparison. The open-source code is on https://github.com/Kid-Liet/Reg-GAN.

**Pix2pix** is a typical supervised GAN using L1 loss functions to enforce pixel-wise similarity between the predicted images and ground-truth images. In the implementation, the generator and discriminator shared the same network architecture as ours for a fair comparison. The open-source code is on https://github.com/junyanz/pytorch-CycleGAN-and-pix2pix.

**CycleGAN** proposes a cycle consistency loss to constrain two generators that are reverse to each other. A cycleGAN consists of two sets of generators and discriminators. In the implementation, the generator and discriminator shared the same network architecture as ours for a fair comparison. The open-source code is on https://github.com/junyanz/pytorchCycleGAN-and-pix2pix.

**UNIT** leverages the assumption that the latent coding space is shared by different modalities for image-to-image translation. The open-source code is on https://github.com/mingyuliutw/UNIT.

**MUNIT** disentangles the representation to a content representation and a style representation for image-to-image translation. The open-source code is on https://github.com/NVlabs/MUNIT.

**NICE-GAN** is a compact and effective network architecture for image-to-image translation, which is achieved by reusing the discriminator for encoding. The open-source code is on https://github.com/alpc91/NICE-GAN-pytorch.

**RegGAN-NC** is a variant of RegGAN (without cycle consistency). It incorporates a registration module to adaptively fit misaligned distribution. Specifically, it consists of a generator, a registration module and a discriminator. The open-source code is on https://github.com/Kid-Liet/Reg-GAN.

**RegGAN-C** is a variant of RegGAN with Cycle consistency. It consists of two generators, two discriminators and one registration module. The open-source code is on https://github.com/Kid-Liet/Reg-GAN.

## B.2. Implementation of DA-GAN

DA-GAN was implemented in PyTorch with Python 3.7. The network architecture of DA-GAN consists of two modality generators, two symmetric spatial aligners, and two deformation-aware discriminators.

**Modality generator** Each modality generator uses 2 downsampling layers, 9 residual blocks, and 2 upsampling layers, following the implementation of Johnson et al. (Johnson et al., 2016). Specifically, we use $Ck$ to denote 7*7 convolution-InstanceNorm-ReLU layer with k filters and stride 1, $Dk$ to denote the downsampling 3*3 convolution-instanceNorm-ReLU layer with k filters and stride 2, $Rk$ to denote a residual block containing two 3*3 convolutional layers with same number of filters, $Uk$ to denote the upsampling 3*3 fractional-strided-convolution-instanceNorm-ReLU layer with k filters and stride 0.5. The architecture of a modality generator is C64-D128-D256-R256-R256-R256-R256-R256-R256-R256-R256-R256-U128-U64-C1.

**Symmetric spatial aligner** Each symmetric spatial aligner consists of 4 transformation regressors and 4 spatial transformer networks. Each transformation regressor uses ResUnet architecture, containing 7 encoder layers, 3 residual blocks, 7 decoder layers, and skip connections, following the implementation of RegGAN (Kong et al., 2021). Specifically, we use $Dk$ to denote downsampling 3*3 convolution-LeakyReLU (LeakyReLU with a slope of 0.2) with k filters and stride 1, $Rk$ to denote a residual block containing two 3*3 convolutional layers, $Uk$ to denote upsampling 3*3 convolution-LeakyReLU (LeakyReLU with a slope of 0.2) with k filters and stride 1. The backbone architecture of the transformation regressor is D32-D64-D64-D64-D64-D64-D64-R64-R64-R64-U64-U64-U64-U64-U64-U32, followed by a refinement layer (a residual block and a 1*1 convolutional layer) and an output layer (3*3 convolution layer).

**Deformation-aware discriminator** For deformation-aware discriminator, it was implemented with a 70*70 PatchGAN (Isola et al., 2017). Specifically, If we denote a 4*4 convolution-instanceNorm-LeakyReLU layer with k filters and stride 2 as $Ck$ (LeakyReLU with a slope of 0.2). The architecture of the discriminator is C64-C128-C256-C512. After the last layer, another convolution is applied to generate 1-dimensional classification results. No instanceNorm is used for the first $C$64 layer.

Table 6: Weights for each loss function in DA-GAN.

| Loss | $L_{sc}$ | | $L_{mic}$ | | | $L_{adv\_da}$ |
|---|---|---|---|---|---|---|
| Weights | $\lambda_{reg}$ | $\lambda_{smt}$ | $\lambda_{ic\_reg}$ | $\lambda_{ic\_gen}$ | $\lambda_{ic\_joint}$ | $\lambda_{adv\_da}$ |
| Values | 20 | 10 | 10 | 10 | 10 | 1 |

Table 7: Ablation study on the proposed losses $L_{mic}$ and $L_{adv\_da}$.

|   | $L_{ic\_reg}$ | $L_{ic\_gen}$ | $L_{ic\_joint}$ | $L_{adv}$ | $L_{adv\_da}$ | MAE3D | PSNR3D | SSIM3D |
|---|---|---|---|---|---|---|---|---|
| A | Y | | | | Y | 38.88±5.86 | 31.78±1.30 | 0.713±0.018 |
| B | | Y | | | Y | 37.23±6.00 | 32.13±1.21 | 0.719±0.021 |
| C | | | Y | | Y | 40.99±7.53 | 31.52±1.35 | 0.709±0.023 |
| D | Y | Y | | | Y | 41.15±7.31 | 31.73±1.31 | 0.712±0.021 |
| E | Y | | Y | | Y | 40.08±7.72 | 31.66±1.22 | 0.712±0.024 |
| F | | Y | Y | | Y | 36.86±7.58 | 32.32±1.43 | 0.720±0.026 |
| G1 | Y | Y | Y | Y | | 40.57±12.41 | 31.72±2.34 | 0.721±0.027 |
| **G2** | **Y** | **Y** | **Y** | | **Y** | **35.86±6.91** | **32.49±1.38** | **0.725±0.024** |

**Loss weights** Lastly, the weights for different losses in DA-GAN were shown in Table 6, which followed RegGAN (Kong et al., 2021) for a fair comparison. Sensitivity analysis in Appendix F shows the model performance is robust across different choices of hyperparameter values.

## Appendix C. Ablation study with complete details

Table 3 shows the results of an ablation study with full details on the multi-objective inverse consistency loss $L_{mic}$ and deformation-aware adversarial loss $L_{adv\_da}$. Firstly, for $L_{mic}$, we experimented on the 7 settings on different combinations of $\{L_{ic\_reg}, L_{ic\_gen}, L_{ic\_joint}\}$. As shown in Table 3, by comparing the settings A-F and G2, we concluded that our proposed $L_{mic}$ that was composed of all three IC losses achieved the best results. Secondly, for $L_{adv\_da}$, we compared DA-GAN using conventional adversarial loss $L_{adv}$ (G1) and deformation-aware adversarial loss $L_{adv\_da}$ (G2). The results show that $L_{adv\_da}$ outperformed its counterpart. Overall, the ablation study demonstrated the effectiveness of both proposed components $L_{mic}$ and $L_{adv\_da}$.

## Appendix D. Convergence analysis during training

Figure 6 shows the validation NMAE during the training process for different levels of non-affine misalignments in the brain dataset. The results demonstrate that DA-GAN successfully converged under different levels of non-affine misalignments.

## Appendix E. Comparison of complexity and accuracy

As shown in Table 8, our model achieves better synthesis accuracy (9.1% increase of MAE3D) at the cost of additional complexity (doubled no. parameters and size of the model) compared with RegGAN-NC. However, we must emphasize that synthesis accuracy is clinically critical in radiotherapy planning because it reduces unnecessary radiation-related toxicity in patients and is vital to patients' safety (Burnet et al., 2004). Further, our model complexity is still smaller than state-of-the-art GANs (e.g., MUNIT and NiceGAN), and our inference time is fast (0.013s fps).

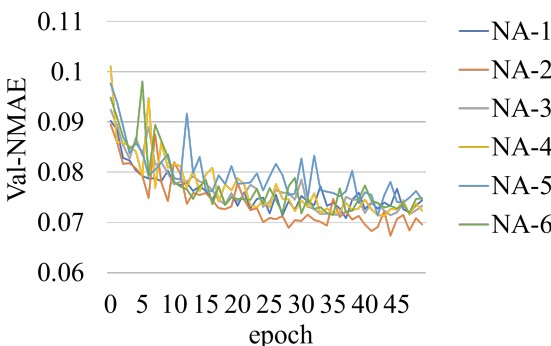

Figure 6: Convergence analysis of DA-GAN on brain dataset with non-affine misalignments.

Table 8: Comparison of complexity and accuracy of the generative models on the lung MRI-CT dataset

| Lung MRI-CT data | MUNIT | NiceGAN | RegGAN-NC | RegGAN-C | Ours |
|---|---|---|---|---|---|
| MAE3D | 39.8 | 42.7 | 39.5 | 40.8 | **35.9** |
| No. Parameters (M) | 44.3 | 112.3 | **16.2** | 30.3 | 36.5 |
| Size of the model (mb) | 169.0 | 428.3 | **61.8** | 115.7 | 139.2 |
| fps (s) | 0.02 | 0.10 | **0.01** | **0.01** | **0.01** |

## Appendix F. Sensitivity analysis on hyperparameters

We conducted a sensitivity analysis on hyperparameters of loss weights on the lung MRI-CT dataset. As shown in Table 9, the results (fluctuating slightly from 0.009 to 0.010) indicate that the model performance is robust across different choices of hyperparameter values.

Table 9: Sensitivity analysis on hyperparameters of loss weights on the lung MRI-CT dataset

| $\lambda_{reg}$ | MAE2D | $\lambda_{smt}$ | MAE2D | $\lambda_{mic}$ | MAE2D |
|---|---|---|---|---|---|
| 5 | 0.009 | 1 | 0.011 | 1 | 0.010 |
| 10 | 0.010 | 5 | 0.010 | 5 | 0.010 |
| 20 | 0.009 | 10 | 0.009 | 10 | 0.009 |
| 30 | 0.010 | 20 | 0.010 | 20 | 0.009 |

## Appendix G. Inverse consistency in registration

Inverse consistency is often implemented by symmetrising cost functions (Christensen and Johnson, 2001) or computing the cost function in "mid-space" (Reuter et al., 2010). A latest work proposed a multi-step IC registration to allow for coarse-to-fine registration

((Greer et al., 2023)). As our work primarily focuses on image synthesis with better-handled misalignment, we design a single-step multi-objective IC loss which jointly optimises the image generation and registration. For efficiency reasons, our solution does not rely on the scaling and squaring technique and its costly numerical integration. The results show that the proposed IC loss significantly contributes to the improved generative performance (p-value $< 0.001$) on misaligned data. In future work, we will investigate the influence of better regularisation methods, such as Total Variation Regularisation (Vishnevskiy et al., 2016).

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
