# OpenReview forum: "Deformation-aware GAN for Medical Image Synthesis with Substantially Misaligned Pairs"
_MIDL.io/2024/Conference — MIDL 2024 Poster_

### Official Review · Reviewer_3gB1 · 2024-02-25

**Confidence:** 5
**Preliminary Rating:** 3
**Final Rating:** 3.5

**Summary:**

The paper works towards improved multi-modal image translation in the presence of spatial misalignment. In contrast to other unpaired work that simple ignores the potential of registration they built upon a recent (2021) NeurIPS paper, which introduced RegGAN for incorporating registration into image translation. The claimed contribution is three-fold: 1) more emphasis on inverse consistency within the registration part 2) a new deformation aware adversarial loss 3) real-world experiments on 3D lung CT-MRI.

**Strengths:**

- The paper is in general well written and the method raises interest.
- I really quite like the deformation aware adversarial loss, which intuitively makes a lot of sense and hasn't been used in prior work by Kong et al. (NeurIPS 2021).
- The reproducibility for the simulation experiments on BRATs is considered good, despite not providing code, but sufficient implementation details in the appendix.
- The evaluation contains both a comparison to several comparable SOTA works (for GAN-based image translation) as well as an ablation study.

**Weaknesses:**

The paper has several weaknesses in its current form:
- The claims on multimodal deformable registration being too complex are extremely overstated. References are only given to two papers from more than 10 years ago (Christensen 2002 and Sotiras 2013) and no new SOTA method for multimodal registration is applied as a comparison, I would not support acceptance if this major weakness is not fixed (see details in detailed recommendations below)
- The private experimental data is neither described in detail (even the Appendix gives no insight of dimensions, voxel spacing etc) nor evaluated with respect to the achieved registration quality (no TRE, no Dice overlap of manual annotations) and even more strikingly the automatic pipeline for pre-registration that was used as initial status is only deemed as "fails to perfectly align images" without any indication what specific errors were observed and whether the proposed method improves upon them
- Given that many very accurate segmentation tools are available for a broad range of anatomical structures for CT (TotalSegmentator) and through domain generalisation / adaptation those can be transferred to MRI the impact of image translation for e.g. radiotherapy planning could be debatable. At least no downstream task performance is discussed, it hence remains unknown whether the achieved PSNR/SSIM is clinically meaningful.

**Detailed Comments:**

The wording for the proposed "multi-level" loss could be slightly misleading because levels are usually used to refer to resolutions and scales of images. Maybe multi-task would be more appropriate. The contributions and improvements compared to RegGAN should be better clarified. On the first impression/reading inverse consistent registration seems key, however Table 3 reveals that the variant F (without inverse consistency, which would be much closer to RegGAN) is only marginally inferior to the proposed one (e.g. PNSR 32.32+-1.43 vs 32.49+-1.38). Is this difference statistically significant? I would suggest a rephrasing that the deformation aware loss is the key element, since this brings an improvement in PSNR of 0.77 (still not huge but probably significant).
As mentioned above, the omission of current SOTA in deformable multimodal registration make the paper currently not acceptable in my opinion. The Learn2Reg 2021 challenge (Hering et al. TMI 2023) contained a similar CT-MRI thorax alignment task, which received several highly accurate (and publicly available) solutions. I would recommend to run both the proposed method on this dataset to obtain an evaluation of target registration error (or its proxy using segmentation overlap) AND use one of the high-performing methods as another baseline for the (paired) image translation part (including the proposed deformation aware loss).
I would be nice to discuss the proposed inverse consistency approach with more mathematically sounded methods, e.g. " Inverse Consistency by Construction for Multistep Deep Registration" MICCAI 2023
Minor comments: Page 6 evaluated in 3D evaluated with MAE3D (duplicate word evaluated)
Caption to Fig. 2 is not informative
Page 2: the statement "mapping leaned by GANS is random within large feasible solution space" is unclear to me

**Justification Of Final Rating:**

While the authors addressed some aspects of my concerns, my overall evaluation of the method remains slightly sceptical. No true SOTA comparison is provided on published results for multi-modal registration, which I believe would have been key to demonstrate the practical gains of the proposed method. Nevertheless the authors promised to follow up on this important aspect in future work (maybe even the possible MELBA journal paper), so I trust they will be able to perform thoroughly if given more time. I now upgraded my score to accept!

**Justification Of The Preliminary Rating:**

While I like the approach and the method comprises novel elements, the experimental validation in particular the improvements over Kong (2021) is a bit underwhelming (if indeed the IC registration loss is not very relevant). Without a proper discussion of newer SOTA work on multimodal registration and its performance in this setting the claimed gains are not necessarily convincing.

**Questions To Address In The Rebuttal:**

- What are the physical details of the real-world CT-MRI data?
- What is the registration accuracy of the pre-alignment and the proposed method based on manual annotations?
- What is the performance of a standard modality translation when employing SOTA multimodal registration?
- What is the significance and/or performance difference of modality translation for downstream radiotherapy tasks compared to multimodal segmentation models?

---

> ### Author Response · Authors · 2024-03-18
> **# Response to Reviewer 3gB1 "Questions To Address in the Rebuttal"**
>
> # Response to Reviewer 3gB1
>
> ## Response to "Questions To Address in the Rebuttal"
>
> > Q1 Physical details of lung CT-MRI data?
>
> Thanks for the reviewer’s valuable feedback. We have added physical details of the lung MRI-CT dataset including voxel
> spacing and dimension in Table 1 below [in AppendixA-Table5].
>
> "For the MRI protocol, both cystic fibrosis and lung cancer cohorts used Siemens 3T scanners with flip angle 5, echo
> time 0.05, repetition time 2.97-3.78. For details of image size and spacing, please refer to Table 5. Both MRI
> and CT were normalised to [-1, 1], cropped to lung regions, resampled to isotropic spacing, and preliminarily
> registered."
>
> Table 1 Physical details of the real-world MRI-CT data
>
> | Lung cancer       | Image size         | No. slice | Pixel spacing         | Slice spacing |
> |-------------------|--------------------|-----------|-----------------------|---------------|
> | 1)MRI-spiral vibe | 320*320            | 176-210   | 1.5*1.5               | 1.5           |
> | 2)CT              | 512*512            | 181-234   | 1.71*1.71             | 2             |
> | **Cystic fibrosis**   | **Image size**         | **No. slice** | **Pixel spacing**         | **Slice spacing** |
> | 1)MRI-spiral vibe | 512\*512 or 416\*416 | 160-241   | 1.1\*1.1 or 1.25\*1.25 | 1.1 or 1.25   |
> | 2)CT              | 512*512            | 267-902   | 0.46\*0.46-0.78\*0.78   | 0.4-1         |
>
> > Q2 Registration accuracy of the pre-alignment and the proposed method based on manual annotation
>
> As suggested, we have added details of the registration accuracy of the pre-alignment and our improvement in Table
> 2 below [Dataset section of the main paper and Appendix A]. The results show registration error (Dice 0.949 on the whole long) in the pre-aligned dataset, while the proposed method improved the alignment to Dice 0.952. Registration errors also occur at bone regions and airways; however, due to the difficulty of annotating on MRI, we will investigate these regions in future work.
>
> Table 2 Registration accuracy based on manual annotation
>
> | |RightLung|LeftLung|Wholelung|
> |---|---|---|---|
> |Pre-alignment(Dice)|0.947|0.950|0.949|
> |CorrField(Dice)|0.940|0.928|0.934|
> |ours(Dice)|0.951|0.953|0.952|
>
> > Q3 Performance of standard image translation with SOTA multimodal registration
>
> As suggested, we have conducted experiments using a SOTA multimodal registration method (CorrField, which ranked 2nd
> best in learn2reg MR-CT registration challenge (Hering et al., 2022)) [ https://grand-challenge.org/algorithms/corrfield/ ] prior to Pix2pix. The results in Table 3 show our method (SSIM 0.731, Dice 0.952) still outperformed CorrField-based method (SSIM 0.708, Dice 0.934) in terms of generative quality (Table 3) and registration error (Table 2). We have added this experiment as an additional baseline [in the Result of the paper].
>
> The results tend to highlight the difficulty of the lung MRI-CT registration challenge. Though CorrField achieved good
> results in thorax-abdomen registration, their task focuses more on the abdomen instead of the thorax (which is more
> challenging). This difficulty is caused by
> - lung MRI has limited image quality due to low proton density,
> - lung MRI-CT involves larger deformation caused by respiratory, heart and body motion.
>
> Our proposed algorithm shows great promise to address this unsolved problem of lung MRI-CT synthesis.
>
> Table 3 Comparison of synthesis results with SOTA registration prior to Pix2pix
>
> | |MAE3D|SSIM3D|PSNR3D|
> |-------------------------|-------------|-------------|------------|
> |CorrField+Pix2pix|48.76±10.87|0.708±0.021|29.55±1.77|
> |Our registration+Pix2pix|39.64±8.31|0.710±0.027|31.88±1.48|
> |The proposed network|35.86±6.97|0.731±0.023|32.49±1.38|
>
> > Q4 Downstream radiotherapy tasks compared with multimodal segmentation.
>
> Thanks for the opportunity to clarify our major downstream application, which is MRI-only radiotherapy (RT) dose
> planning (removing the need of scanning CT) (Spadea et al., 2021), rather than segmentation using multimodal segmentation.
> Specifically,
>
> - MRI-only radiotherapy dose planning is an important clinical application because 1) it decreases the exposure to
>   ionizing radiation for repeated scanning and radiosensitive population, 2) simplifies the clinical workflow, 3)
>   provides more accurate boundaries of soft-tissue organs compared with CT.
> - To demonstrate clinical relevance, our method (MAE 35.86 HU) outperformed the state-of-the-art MRI-only RT
>   application (MAE 54.9 HU) by (Lenkowicz et al., 2022, Radiotherapy and Oncology).
> - We are collecting more data, and will further investigate dose metrics and in-depth analysis of clinical benefits in
>   future clinical publication.
>
> **Reference**
> 1. Hering et al., 2022: https://ieeexplore.ieee.org/abstract/document/9925717
> 2. Spadea et al., 2021: https://aapm.onlinelibrary.wiley.com/doi/full/10.1002/mp.15150
> 3. Lenkowicz et al., 2022: https://www.sciencedirect.com/science/article/pii/S0167814022042608

---

> > ### Author Response · Authors · 2024-03-18
> > **# Response to Reviewer 3gB1  "Weakness"**
> >
> > # Response to Reviewer 3gB1
> >
> > ## Response to "Weakness"
> >
> > > W1 No new SOTA method for multimodal registration as a comparison.
> >
> > As suggested, we conducted experiments using SOTA multimodal registration (CorrField, which ranked 2nd best in
> > learn2reg challenge (Hering et al., 2022)) prior to Pix2pix. The results show our method (SSIM 0.731, Dice 0.952)
> > still outperformed CorrField-based method (SSIM 0.708, Dice 0.934) in terms of generative quality (
> > Table 1) and registration error (Table 2). We have included this in the paper as an additional baseline.
> >
> > The results tend to highlight the difficulty of the lung MRI-CT registration challenge. Though CorrField achieved good
> > results in thorax-abdomen registration, their task focuses more on the abdomen instead of the thorax (which is more
> > challenging).
> >
> > - Specifically, lung MRI-CT registration is more challenging because 1) the image quality of lung MRI is limited due to
> >   low proton density and motion artefacts, and 2) large deformation between images due to a combination of respiratory
> >   motion, heart motion, and body motion.
> > - Due to the significant non-linear deformation that can occur during lung imaging and the challenges in consistently
> >   achieving accurate diffeomorphic alignment, we choose a more resilient registration pipeline comprising Elastix +
> >   anatomy guided + SyN from the ANTs library.
> > - Our proposed algorithm shows great promise to address this unsolved problem of lung MRI-CT synthesis.
> >
> > Table 1 Comparison with SOTA registration prior to Pix2pix
> >
> > | |MAE3D|SSIM3D|PSNR3D|
> > |-------------------------|-------------|-------------|------------|
> > |CorrField+Pix2pix|48.76±10.87|0.708±0.021|29.55±1.77|
> > |Our registration+Pix2pix|39.64±8.31|0.710±0.027|31.88±1.48|
> > |The proposed network|35.86±6.97|0.731±0.023|32.49±1.38|
> >
> > Table 2 Registration accuracy based on manual annotation
> >
> > | |RightLung|LeftLung|Wholelung|
> > |---|---|---|---|
> > |Pre-alignment(Dice)|0.947|0.950|0.949|
> > |CorrField(Dice)|0.940|0.928|0.934|
> > |our network(Dice)|0.951|0.953|0.952|
> >
> > > W2 Private dataset is not described in detail. No error for registration. Whether the proposed method improves upon them.
> >
> > Thanks for the reviewer’s valuable feedback. As suggested, we have added 1) a detailed description of the private
> > dataset in Table 3; 2) details of registration quality of the pre-registration stage; and 3) details of the improvement
> > of registration error.
> >
> > 1. "For the MRI protocol, both cystic fibrosis and lung cancer cohorts used Siemens 3T scanners with flip angle 5, echo
> >    time 0.05, repetition time 2.97-3.78. For details of image size and spacing, please refer to Table 5. Both
> >    MRI and CT were normalised to [-1, 1], cropped to lung regions, resampled to isotropic spacing, and preliminarily
> >    registered."
> >
> > Table 3 Physical details of the real-world MRI-CT data
> >
> > |Lungcancer|Imagesize|No.slice|Pixelspacing|Slicespacing|
> > |-------------------|--------------------|-----------|-----------------------|---------------|
> > |1)MRI-spiralvibe|320*320|176-210|1.5*1.5|1.5|
> > |2)CT|512*512|181-234|1.71*1.71|2|
> > |**Cysticfibrosis**|**Imagesize**|**No.slice**|**Pixelspacing**|**Slicespacing**|
> > |1)MRI-spiralvibe|512\*512or416\*416|160-241|1.1\*1.1or1.25\*1.25|1.1or1.25|
> > |2)CT|512*512|267-902|0.46\*0.46-0.78\*0.78|0.4-1|
> >
> >
> > 2. We summarised the registration quality using Dice overlap of manual annotations on lung regions as shown in Table 2.
> >    The results show that there exists registration error in pre-registration pipeline (Dice 0.949 on whole
> >    lungs), and the proposed method improves the previous registration error (Dice 0.952).
> >
> > 3. Registration errors also occur at bones regions, vessels and airways; however, due to
> >    the difficulty of segmenting these components in lung MRI, we used lungs as an example but will investigate these
> >    regions for future work.
> >
> > > W3 No downstream task is discussed, it hence remains unknown whether achieved PSNR,SSIM is clinically meaningful.
> >
> > Thanks for the opportunity to clarify our major downstream application, which is MRI-only radiotherapy dose planning (
> > removing the need of scanning CT) (Spadea et al., 2021), rather than segmentation using multimodal segmentation.
> > Specifically,
> >
> > - MRI-only radiotherapy dose planning is an important clinical application because 1) it decreases the exposure to
> >   ionizing radiation for repeated scanning and radiosensitive population, 2) it simplifies the clinical workflow, 3)
> >   providing more accurate boundaries of soft-tissue organs compared with CT.
> > - To demonstrate clinical relevance, our method (MAE 35.86 HU) outperformed the state-of-the-art MRI-only RT dose planning
> >   application (MAE 54.9 HU) by Lenkowicz et al., 2022.
> > - We are collecting more data, and will further investigate dose metrics and in-depth analysis of clinical benefits in a
> >   future clinical publication.

---

> > > ### Author Response · Authors · 2024-03-18
> > > **# Response to Reviewer 3gB1  "Detailed comments C1-C3"**
> > >
> > > # Response to Reviewer 3gB1
> > >
> > > ## Response to "Detailed comments"
> > >
> > > > C1 Wording for "multi-level"
> > >
> > > We agree with the reviewer’s comments and have revised the name to “multi-objective”.
> > >
> > > > C2 The contribution of multi-objective inverse consistency (MIC) loss.
> > >
> > > Thanks for the reviewer’s constructive comments. We would like to clarify that both inverse consistency loss and
> > > deformation-aware adversarial loss make statistically significant contributions(p-value<0.001) [in Results].
> > >
> > > 1. We would like to clarify that variant F still consists of components for IC registration, including joint-level IC
> > >    loss and symmetric spatial aligners. Instead, variant B is the most suitable approximation to demonstrate the
> > >    contribution of DA-adversarial loss.
> > > 2. As shown in Table 1 below, DA-adversarial loss (Variant B) improves the baseline RegGAN-C by **0.41 PSNR-3D**. By
> > >    comparing the variant G2 to variant B, MIC loss further improves **PSNR by 0.36**. To summarise, both losses
> > >    contribute greatly to the final performance.
> > > 3. The results of paired t-test show the improvement of both losses on MAE2D, PSNR2D, SSIM2D are statistically
> > >    significant (p-value<0.001). Due to the limited number of 3D volumes, the testing did not show statistical
> > >    significance for 3D metrics (p ~0.15).
> > > 4. Lastly, we would like to emphasize the importance of inverse consistency registration because the DA adversarial loss
> > >    was built upon the output of IC registration y ◦ ϕ(y)  and x ◦ ϕ(x). As a result, it is difficult to fully single out
> > >    the contribution of DA adversarial loss because it is interconnected with IC registration.
> > >
> > > Table 1 Contributions of our two losses
> > >
> > > | |MAE2D|PSNR2D|SSIM2D|MAE3D|PSNR3D|SSIM3D|
> > > |--------------------------------|-----|------|------|------|-------|-------|
> > > |RegGAN-C|0.011|70.23|0.918|40.77|31.72|0.710|
> > > |Variant-B(L_adv_da)|0.010|74.82|0.925|37.23|32.13|0.719|
> > > |Variant-F(L_adv_da+Lic_joint)|0.010|75.58|0.922|36.86|32.32|0.720|
> > > |Variant-G2(L_adv_da+Lmic)|0.009|76.36|0.929|35.86|32.49|0.725|
> > >
> > > > C3 1) The omission of SOTA multimodal registration. 2) Suggestion to run a comparison with SOTA registration.
> > >
> > > As suggested, we conducted experiments using a state-of-the-art multimodal deformable registration method (CorrField,
> > > which ranked second best in the learn2reg challenge) prior to Pix2pix. The results show that our method still
> > > outperformed the SOTA method.
> > >
> > > - Specifically, we used the original implementation of CorrField used in the learn2reg challenge. The results showed
> > >   that our proposed network (SSIM3D 0.731) outperformed CorrField+Pix2pix (SSIM3D 0.708). We also conducted additional
> > >   comparisons of the registration error between our pipeline and CorrField and found our pipeline (whole lung Dice
> > >   0.952) achieved lower error compared with CorrField (whole lung Dice 0.934) in Table 2.
> > >
> > > - We believe that the new baseline experiments with SOTA registration have validated our proposed solution for the
> > >   image-to-image translation task. In terms of registration performance, we also conducted quantitative comparison with
> > >   SOTA registration algorithms on our dataset, though our algorithm is purposed for image synthesis. Due to limited
> > >   rebuttal time and a large number of experiments to run, we will include this further investigation on another new
> > >   registration dataset (learn2reg) in future work.
> > >
> > > - The results tend to highlight the difficulty of the lung MRI-CT challenge. Though algorithms achieve good results in
> > >   thorax-abdomen registration, this task focuses more on the abdomen instead of the thorax. Lung MRI-CT registration is more
> > >   difficult because 1) the image quality of lung MRI is limited due to low proton density and motion artefacts, and 2) large
> > >   deformation between images due to a combination of respiratory motion, heart motion, and body motion. In this
> > >   challenging lung MRI-CT application, multimodal registration has not been thoroughly investigated, while our proposed
> > >   algorithm shows great promise in achieving good synthesis performance.
> > >
> > > Table 2 Registration accuracy based on manual annotation
> > >
> > > | |RightLung|LeftLung|Wholelung|
> > > |---|---|---|---|
> > > |Pre-alignment(Dice)|0.947|0.950|0.949|
> > > |CorrField(Dice)|0.940|0.928|0.934|
> > > |our network(Dice)|0.951|0.953|0.952|
> > >
> > > Table 3 Comparison with SOTA registration prior to Pix2pix
> > >
> > > | |MAE3D|SSIM3D|PSNR3D|
> > > |-------------------------|-------------|-------------|------------|
> > > |CorrField+Pix2pix|48.76±10.87|0.708±0.021|29.55±1.77|
> > > |Our registration+Pix2pix|39.64±8.31|0.710±0.027|31.88±1.48|
> > > |The proposed network|35.86±6.97|0.731±0.023|32.49±1.38|

---

> > ### Comment · Reviewer_3gB1 · 2024-03-22
> >
> > I thank the authors for several clarifications and their attempt to include a quality assessment of the achieved multi-modal registration. I am, however, not very much convinced by the additional experiment and would not recommend to include it. Assessing registration quality only for lungs (very large structures) is not very meaningful and it was directly mentioned that (mis)alignment of vertebra etc. would be more informative. Furthermore it is a bit strange to only pick the second best Learn2Reg method as a comparison and in addition not do the test the other way around: apply the proposed method to the Learn2Reg dataset. The comparisons with the obtained registration of the proposed method + Pix2Pix compared to their full pipeline in terms of MAE3d/PSNR (Table 3) is of interest as it demonstrates the influence of (a good) alignment on translation. I think the other questions were not answered directly, but could see that at least the caption of Fig. 2 was amended in the revised paper. I therefore stick to my initial rating.

---

> ### Author Response · Authors · 2024-03-18
> **# Response to Reviewer 3gB1 "Detailed comments C4 and minor comments" and "Justification of Rating"**
>
> > C4 Discuss the proposed inverse consistency approach with more mathematically sounded methods
>
> As suggested, we have added a discussion and comparison with other IC approaches (including the suggested
> reference)  [in Appendix G] and as below.
>
> “Inverse consistency is often implemented by symmetrising cost functions (Christensen & Johnson, 2001) or computing the
> cost function in "mid-space” (Reuter et al., 2010). A recent work proposed a multi-step IC registration to allow for
> coarse-to-fine registration (Greer et al., 2023). As our work primarily focuses on image synthesis with better-handled
> misalignment, we design a single-step multi-objective IC loss which jointly optimises the image generation and
> registration. For efficiency reasons, our solution does not rely on the scaling and squaring technique and its costly
> numerical integration. The results showed the proposed IC loss significantly contributed to the improved generative
> performance (p-value < 0.001) on misaligned data.”
>
>
> > Minor 1: Page 6 evaluated in 3D evaluated with MAE3D (duplicate word evaluated)
>
> As suggested, we have removed the duplicate words.
>
> > Minor 2: Caption to Fig. 2 is not informative
>
> As suggested, we will add an informative description to the Fig. 2 caption in the manuscript as below:
> (a) Network architecture of DA-GAN. (b) Lmic loss dynamically enhances image correspondence from three objectives. (c)
> Ladv_da loss guides discriminators to learn deformation for improved image fidelity.
>
> > Minor 3:Page 2: the statement "mapping leaned by GANS is random within large feasible solution space" is unclear to me.
>
> Thanks for the reviewer’s comment and apologies for the unclear English. We would like to clarify that according to Shen
> et al (Shen et al., 2020), cycle consistency mapping used in unsupervised GANs is still not strictly one-to-one mapping,
> which is an important condition in intra-subject medical image synthesis.
>
> ## Response to "Justification of the preliminary rating"
>
> > While I like the approach and the method comprises novel elements, the experimental validation in particular the improvements over Kong (2021) is a bit underwhelming (if indeed the IC registration loss is not very relevant). Without a proper discussion of newer SOTA work on multimodal registration and its performance in this setting the claimed gains are not necessarily convincing.
>
> We would like to express our gratitude for the reviewer’s valuable comments and the opportunity to address the concerns
> regarding 1) IC-registration loss, and 2) comparison with methods based on SOTA multimodal registration.
>
> - Firstly, we would like to clarify that the IC registration loss is highly relevant as it shows statistically
>   significant improvement over our variant without IC registration loss on MAE2D, PSNR2D, SSIM2D (p-value<0.001). Also,
>   experiments show the improvement of DA adversarial loss is statistically significant as well (p-value<0.001) over
>   Kong (2021). As a result, we believe that both of our novel losses make statistically significant contributions,
>   and our improvement over Kong (2021) is statistically significant as well.
> - Secondly, we compared our approach with methods based on SOTA multimodal registration (CorrField, second best in the
>   learn2reg challenge). The result demonstrated that our approach outperformed the CorrField-based approach on both
>   synthesis quality (SSIM 0.708) and registration errors (Dice 0.952 on the whole lungs), compared with CorrField-based
>   approach (SSIM 0.731, Dice 0.934). We believe that this additional baseline experiment in response to your feedback
>   has strengthened the credibility and relevance of our study.
>
> Further, we have provided point-to-point responses to Weakness, Detailed comments and Rebuttal Questions and revised them
> in the manuscript (Colored in Brown). Please find a summary as below:
>
> - our IC registration loss is a statistically significant improvement (p-value<0.001) _[in Results]_;
> - our performance over Kong (2021) is statistically significant (p-value<0.001) _[in Results]_;
> - our work outperformed SOTA multimodal registration (CorrField) prior to pix2pix in terms of both generative quality,
>   and registration alignment _[in Results]_.
> - We provided more physical details of the lung MRI-CT dataset _[in Appendix A]_.
> - We provided registration accuracy of pre-alignment and the improvement by our method _[in Appendix A]_.
> - We clarified the our major downstream clinical application is MRI-only radiotherapy dose planning (removing the need of CT scanning),
>   instead of multimodal segmentation _[in Introduction]_.
> - We have further discussed the comparison with mathematically sound IC approaches _[in AppendixG]_.

---

> ### Author Response · Authors · 2024-03-26
>
> Thanks for the reviewer’s assessment and advice. We would like to reiterate that our primary contribution lies in the development of a novel medical image-to-image (I2I) translation model. Our investigation highlighted the limitation of existing I2I methods when dealing with pairs of images affected by significant non-linear deformations.
> - Specifically, we discovered that current architectures, such as RegGAN, are still affected by substantial misalignment, and could benefit greatly from our proposed network (with a novel inverse-consistency loss for improved mapping correspondence and a deformation-aware adversarial loss for optimised image fidelity).
> - We validated our I2I method on the benchmark dataset used in RegGAN 2021 and further on a clinical lung MRI-CT dataset, compared with SOTA GAN methods. The results show statistically significant improvement compared with baseline RegGAN (p-value<0.001).
>
>
> In light of time constraints, we opted to select the "second best method" in learn2reg challenge, primarily because it
> provides easy-to-access docker container to facilitate reproducibility, while the best solution  ConvexAdam does not. It's worth noting the minimal discrepancy with the best-performing method:
> CorrField even outperformed ConvexAdam in 2 out of 5 registration metrics, including Dice and HD95, though overall registration performance (0.81+/-0.02) is only marginally below ConvexAdam (0.82+/-0.01).
>
>
> While we acknowledge the relevance of utilising the Learn2Reg registration dataset, particularly if our contribution was solely focused on proposing a new registration method, we firmly believe that the task of image synthesis holds greater significance for the targeted downstream applications (e.g., MR-only radiotherapy planning). Additionally, the current manuscript already contains a substantial amount of experimental content, so we plan to explore the registration capabilities of our approach in future work.

---

### Official Review · Reviewer_tPyS · 2024-02-28

**Confidence:** 5
**Preliminary Rating:** 4
**Recommendation:** Poster

**Summary:**

This paper adresses medical image translation in the context of "not so well aligned" patient image pairs, what I would call the “gray zone” i.e. where reconstruction loss-based models (e.g. pix2pix) cannot be used due to excess registration errors but where fully unsupervised models (e.g. cycleGAN) would discard too much of relevant available information. The most significant work previously addressing this issue is the RegGAN model from NeurIPS 2021 [Kong et al.], where a spatial transformer is included in both pix2pix and cyclegan-like models.

The model comprises 1) a “symmetric registration” loss, a 2) “multi inverse consistency” loss and a 3) “deformation aware” adversarial loss

1) The symmetric loss learns a deformation field (DVF) $\phi$ between source and target and another one between target and source and ensures that the synthetic image and the target image correspond to each other by mapping both source using a $\ell_1$ cost. Also, a $\ell_2$ regularizer is added to enforce the smoothness of the field, as it is often the case in image registration.

2) inverse consistency makes sure that forward and backward mappers correspond to each other by performing multiple cyclic losses at the image and the DVF level (3 loss terms)

3) the deformation aware adversarial term jointly discriminates the synthetic image and the warped synthetic image.

[Kong et al.] https://doi.org/10.48550/arXiv.2110.06465

**Strengths:**

First, structure and language of the paper are appropriate, the paper follows general scientific principles, and mentions prior work (in particular RegGAN, Kong et al., to which it is compared)

The paper is technically sound and the results are convincing. This "gray zone" is not enough tackled in medical imaging, so it is welcome to see a paper that tries to push a bit further the idea of supervised I2I even in the case of misalignments.

The ablation study is also much welcome given the number of loss terms to be optimized by the network. From this, the most decisive contribution seems to be the "deformation aware adversarial loss", i.e. the training of the discriminator both using the warped and unwarped images, with a significant gain in performance with this discriminator over a more conventional image based discriminator.

**Weaknesses:**

The main competitor, RegGAN, upon which this method is obviously built, is not enough discussed in the paper. The only mention in the text (apart from comparative evaluation) does not address to what extent the work is different. Essentially, in my opinion, this work is a refinement / extension of RegGAN with interesting contributions at the cost of additional complexity. This should be further clarified in the text. At the moment, it seems that the work is presented as novel in its intention to tackle the weakly aligned problem, which is not the case given prior works on it.

Many terms have to be  jointly optimized, making the method quite complex. Of course, these terms come with hyperparameters that need to be tuned. The loss weights have been chosen according to RegGAN (for fairness) which is fine. However this complexity is not discussed enough in the paper.

Although not absolutely necessary, it would have been nice to also include results using pix2pix with non-rigidly registered images prior to I2I, to better emphasize the interest of the joint optimization problem tackled.

**Detailed Comments:**

Further comments:
 I think that the name of the method is inappropriate. SAGAN is a very famous, historical GAN model with 4000+ cites [Zhang et al. 2018]. This is confusing to see a "SA-GAN" here in a very similar context. I would encourage you to change the name of the model. What about something that reminds the filiation with RegGAN upon which this work is heavily relying ?

The $\ell_2$ smoothness imposed on the field may also reduce its ability to achieve realistic deformation, especially in case of large deformation. This is especially true in the lung where better regularizers can achieve much  better results (see e.g. [Vishnevskiy et al. TMI 2017] .  This could be mentioned.


[Zhang et al. 2018] https://doi.org/10.48550/arXiv.1805.08318

**Justification Of The Preliminary Rating:**

This is a sound paper addressing an important problem, whose positioning in terms of methodological contribution with respect to Kong et al. 21 is not sufficiently clear. The paper is however very fluent and sound. I am confident that this aspect can be better addressed post rebuttal.

**Questions To Address In The Rebuttal:**

- clarification of the contribution with respect to RegGAN-NC model, especially regarding the added complexity
- mention of more recent, better models for unsupervised/supervised I2I, such as e.g. transformers or diffusion-based models (this is old state of the art)
- name change

---

> ### Author Response · Authors · 2024-03-18
> **Response to Reviewer tPyS "Questions To Address in the rebuttal"**
>
> # Response to Reviewer tPyS
>
> ## Response to "Questions To Address in the Rebuttal"
>
> > Q1 Clarification of the contribution with respect to RegGAN-NC model, especially regarding the added complexity
>
> Thanks for the reviewer’s valuable comments. In the manuscript, we have clarified 1) methodological
> contributions [in Introduction], 2)
> analytical contribution with RegGAN regarding the improved accuracy (that is clinically critical) and additional
> complexity in Table 1 [Appendix D], and 3) clarified that we are addressing two specific challenges (suboptimal data
> mapping and degraded image fidelity) when multimodal pairs are substantially misaligned that are not addressed by
> RegGAN [in Abstract].
>
> 1. Methodological contribution with baseline RegGAN.
>     - Compared with the directional registration module in RegGAN, we proposed a **multi-objective inverse-consistency
>       loss** to enforce bidirectional symmetric registration. This aims to enhance dynamic learning of unique and
>       one-to-one mapping when the real-world setting involves large deformation between pairs.
>     - Compared with the standard patch discriminator in RegGAN, we proposed a **deformation-aware adversarial loss** to
>       disentangle the mismatched spatial morphology from the judgement of image fidelity.
>     - In terms of network architecture, we designed **symmetric spatial aligners** in our network (compared with a
>       single registration module in RegGAN) to enable inverse-consistency registration and deformation-aware adversarial
>       loss.
> 2. Analytical contribution with baseline RegGAN.
>     - In terms of **accuracy and complexity**, as shown in Table 1 below, our model achieves better synthesis accuracy (
>       9.1% increase of MAE3D) at the cost of additional complexity (doubled no. parameters and size of the model)
>       compared with RegGAN-NC. However, synthesis accuracy is clinically critical in radiotherapy planning because it
>       reduces unnecessary radiation-related toxicity in patients and is vital to patients’ safety (
>       Burnet et al., 2004). Further, our model complexity is still smaller than state-of-the-art GANs (e.g., MUNIT and
>       NiceGAN), and our inference time is fast (0.013s fps).
>     - We validated the methods on a **real-world lung MRI-CT dataset** beside the simulated BraTS dataset used in
>       RegGAN.
> 3. In terms of “challenges”, we have clarified in the manuscript that we are addressing two specific weakly-aligned
>    challenges that are not addressed by RegGAN. Specifically, the challenges include (1)
>    suboptimal data mapping caused by correspondence ambiguity, and (2) degraded image fidelity caused by the influence
>    of spatial morphology on the discriminator.
>
>     - To address the suboptimal data mapping, we propose a multi-objective inverse consistency loss to enforce
>       bidirectional symmetric registration, which aims to enhance dynamic learning of unique and one-to-one mapping when
>       real-world setting involves large deformation between pairs.
>     - To address the degraded image fidelity, we propose a deformation-aware adversarial loss to disentangle the
>       mismatched spatial morphology from the judgement of image fidelity.
>
> Table 1. Comparison of complexity and accuracy.
>
> |Methods|MUNIT|NiceGAN|RegGAN-NC|RegGAN-C|Ours|
> |----------------------|-----|-------|---------|--------|--------|
> |MAE3D-lung|39.8|42.7|39.5|40.8|**35.9**|
> |No.Parameters(M)|44.3|112.3|**16.2**|30.3|36.5|
> |Size of the model(mb)|169.0|428.3|**61.8**|115.7|139.2|
> |fps(s)|0.02|0.10|**0.01**|**0.01**|**0.01**|
>
> > Q2 Mention of more recent models of I2I
>
> As suggested, we have included the latest I2I diffusion models (DDPM (Ho et al, 2020), and DDIM (Song et al, 2021)) in
> the manuscript as below. Further, we have conducted an experimental comparison with diffusion models. The results show
> our method (SSIM 0.731) outperformed DDPM (SSIM 0.592) and DDIM (SSIM 0.590) on the lung MRI-CT dataset. As diffusion
> models are data-hungry to train, we suspected that the limited performance of diffusion models was related to the small
> data size and will further investigate the potential to extend our work to diffusion models as a future work.
>
> [Introduction] "Diffusion models have shown great potential in computer vision applications due to their strength in
> capturing distributions; however, they are computationally expensive and data-hungry to train, hindering their
> application in the medical domain."
>
> > Q3 Name change
>
> As suggested, we have renamed the proposed network as Deformation-aware GAN.
>
> **Reference**
>
> 1. Ho, J., Jain, A., & Abbeel, P. (2020). Denoising diffusion probabilistic models. Advances in Neural Information
>    Processing Systems, 33, 6840–6851.
> 2. Song, J., Meng, C., & Ermon, S. (2021). Denoising diffusion implicit models. ICLR Conference.

---

> > ### Author Response · Authors · 2024-03-18
> > **# Response to Reviewer tPyS "Weakness"**
> >
> > # Response to Reviewer tPyS
> >
> > ## Response to "Weakness"
> >
> > > Q1 1) Comparison with the main competitor RegGAN, 2) clarify contribution at the cost of complexity, 3) clarify intention and challenges.
> >
> > Thanks for the reviewer’s valuable comments. In the manuscript, we have clarified 1) methodological
> > contributions [in Introduction], 2)
> > analytical contribution with RegGAN regarding the improved accuracy (that is clinically critical) and additional
> > complexity in Table 1 [Appendix D], and 3) clarified that we are addressing two specific challenges (suboptimal data
> > mapping and degraded image fidelity) when multimodal pairs are substantially misaligned, which are not addressed by
> > RegGAN [in Abstract].
> >
> > 1. Methodological contribution with baseline RegGAN.
> >     - Compared with the directional registration module in RegGAN, we proposed a **multi-objective inverse-consistency
> >       loss**
> >       to enforce bidirectional symmetric registration, which aims to enhance dynamic learning of unique and one-to-one
> >       mapping when real-world setting involves large deformation between pairs.
> >     - Compared with the standard patch discriminator in RegGAN, we proposed a **deformation-aware adversarial loss** to
> >       disentangle the mismatched spatial morphology from the judgement of image fidelity.
> >     - In terms of network architecture, we designed **symmetric spatial aligners** in our network (compared with a
> >       single registration module in RegGAN) to enable inverse-consistency registration and deformation-aware adversarial
> >       loss.
> > 2. Analytical contribution with baseline RegGAN.
> >     - In terms of **accuracy and complexity**, as shown in Table 1 below, our model achieves better synthesis accuracy (
> >       9.1% increase of MAE3D) at the cost of additional complexity (doubled no. parameters and size of the model)
> >       compared with RegGAN-NC. However, synthesis accuracy is clinically critical in radiotherapy planning because it
> >       reduces unnecessary radiation-related toxicity in patients and is vital to patients’ safety (
> >       Burnet et al., 2004). Further, our model complexity is still smaller than state-of-the-art GANs (e.g., MUNIT and
> >       NiceGAN), and our inference time is fast (0.013s fps).
> >     - We validated the methods on a **real-world lung MRI-CT dataset** beside the simulated BraTS dataset used in
> >       RegGAN.
> > 3. In terms of “challenges”, we have clarified in the manuscript that we are addressing two specific weakly-aligned
> >    challenges that are not addressed by RegGAN. Specifically, the challenges include (1)
> >    suboptimal data mapping caused by correspondence ambiguity, and (2) degraded image fidelity caused by the spatial
> >    morphology on the discriminator.
> >
> >     - To address the suboptimal data mapping, we propose a multi-objective inverse consistency loss to enforce
> >       bidirectional symmetric registration, which aims to enhance dynamic learning of unique and one-to-one mapping when
> >       real-world setting involves large deformation between pairs.
> >     - To address the degraded image fidelity, we propose a deformation-aware adversarial loss to disentangle the
> >       mismatched spatial morphology from the judgement of image fidelity.
> >
> > Table 1. Comparison of complexity and accuracy.
> >
> > |Methods|MUNIT|NiceGAN|RegGAN-NC|RegGAN-C|Ours|
> > |----------------------|-----|-------|---------|--------|--------|
> > |MAE3D-lung|39.8|42.7|39.5|40.8|**35.9**|
> > |No.Parameters(M)|44.3|112.3|**16.2**|30.3|36.5|
> > |Size of the model(mb)|169.0|428.3|**61.8**|115.7|139.2|
> > |fps(s)|0.02|0.10|**0.01**|**0.01**|**0.01**|
> >
> > > W2 Hyperparameter complexity
> >
> > Thanks for the reviewer’s insightful comment. We have conducted a sensitivity analysis to investigate the influence of
> > hyperparameters on the lung MRI-CT dataset [in Appendix E]. In Table 2 below, results show MAE fluctuated slightly from
> > 0.009 to 0.010 for different loss weights, which indicates that the model performance is robust across different choices
> > of hyperparameter values.
> >
> > Table 2 Sensitivity analysis on hyperparameters of loss weights
> >
> > | lambda_reg | MAE2D | lambda_smt | MAE2D | lambda_ic | MAE2D |
> > |-------------|-------|------------|-------|------------|-------|
> > | **5**  | 0.009 | **1**  | 0.011 | **1** | 0.010 |
> > | **10**  | 0.010 | **5** | 0.010 | **5** | 0.010 |
> > | **20**  | 0.009 | **10** | 0.009 | **10** | 0.009 |
> > | **30**  | 0.010 | **20** | 0.010 | **20** | 0.009 |
> >
> > > W3 Comparison with pix2pix with registered images prior to I2I
> >
> > Thanks for the reviewer’ valuable comments. Firstly, I would like to clarify that a non-rigid registration pipeline has
> > been used to initially align images before pix2pix on lung MRI-CT dataset. The registration pipeline includes elastix
> > registration (SimpleElastix library), structure-guided registration (demon registration from ANTs library) and
> > antsRegistrationSyNQuick (ANTs library). The results show our method outperformed pix2pix with non-rigidly registered
> > images prior to I2I.

---

> > > ### Author Response · Authors · 2024-03-18
> > > **# Response to Reviewer tPyS (Response for Detailed comments and Justification of Rating)**
> > >
> > > # Response to Reviewer tPyS (Response for Detailed comments and Justification)
> > >
> > > ## Response to "Detailed comments"
> > >
> > > > D1 The name of the model conflicts with self-attention GAN
> > >
> > > As suggested, we will rename the proposed network as Deformation-aware GAN (DA-GAN).
> > >
> > > > D2 Better regularisation can achieve better results compared with l2 smoothness.
> > >
> > > Thanks for the reviewer's insightful comments. As suggested, we will further investigate the influence of better
> > > regularisation methods such as Total Variation Regularisation (Vishnevskiy et al., 2016) to the proposed network as our
> > > future work and will mention this in our manuscript.
> > >
> > > “As a future work, we will investigate the influence of better regularisation methods such as Total Variation
> > > Regularisation (Vishnevskiy et al., 2016) to the proposed network.”
> > >
> > >
> > > ## Response to "Justification of The Preliminary Rating"
> > >
> > > > This is a sound paper addressing an important problem, whose positioning in terms of methodological contribution with respect to Kong et al. 21 is not sufficiently clear. The paper is however very fluent and sound. I am confident that this aspect can be better addressed post rebuttal.
> > >
> > > Thanks for the reviewer’s valuable comments and trust. We have further clarified the methodological contributions, and added
> > > comprehensive comparisons with Kong et al.’s RegGAN (especially regarding improved synthesis accuracy that is clinically
> > > critical and added complexity). Accordingly, we have incorporated the clarification into the abstract and introduction.
> > >
> > > Also, we have provided point-to-point responses to Weakness, Detailed comments and Rebuttal Questions and revised them
> > > in the manuscript (Colored in Blue). Please find a summary as below:
> > >
> > > - the methodological and analytical contribution compared with RegGAN _[in Abstract and Introduction of the manuscript]_
> > >   , especially regarding improved accuracy (which is clinically important) and additional complexity _[Appendix D]_.
> > > - We compared our method with diffusion models _[in Introduction]_.
> > > - We changed the model names to Deformation-aware GAN to avoid possible confusion with self-attention GAN.
> > > - We conducted a sensitivity analysis of our model's hyperparameters _[in Appendix E]_.
> > >
> > >
> > > **Reference**
> > > 1. Vishnevskiy, V., Gass, T., Szekely, G., Tanner, C., & Goksel, O. (2016). Isotropic total variation regularization of
> > >     displacements in parametric image registration. IEEE Transactions on Medical Imaging, 36(2), 385–395.

---

### Official Review · Reviewer_DaaA · 2024-02-28

**Confidence:** 4
**Preliminary Rating:** 5
**Final Rating:** 5

**Summary:**

This paper discusses a self-aligning GAN model proposed for medical image synthesis, addressing challenges with substantially misaligned image pairs.

The model incorporates multi-level inverse consistency and deformation-aware adversarial loss to dynamically correct misalignment during image synthesis.

Experimental results demonstrate the superior performance of the proposed SA-GAN on both simulated and real-world datasets, indicating its potential for various medical image synthesis tasks.

The model's network architecture includes modality generators, symmetric spatial aligners, and deformation-aware discriminators to achieve dynamic alignment and improve image fidelity.

The proposed SA-GAN shows promise for applications such as radiotherapy treatment planning by effectively utilizing misaligned image pairs.

**Strengths:**

The strengths of the paper "Self-aligning GAN for Medical Image Synthesis with Substantially Misaligned Pairs" lie in its innovative approach to addressing the challenges of medical image synthesis with substantially misaligned pairs. The proposed self-aligning GAN (SA-GAN) introduces two new loss functions to generate high-fidelity images while adaptively learning correspondence via symmetric registration. This approach is valuable as it effectively corrects misalignment during image synthesis, which is crucial for tasks such as radiotherapy treatment planning.

The scientific merit of the paper is high due to its comprehensive validation on both simulated and real-world datasets, demonstrating the potential for generalizable medical image synthesis with limited data for clinical applications. The paper's novelty is evident in its approach to dynamically correcting misalignment during image synthesis, which is a significant contribution to the field of medical imaging.

The paper also adequately addresses prior work, providing a comprehensive review of related literature and positioning the proposed SA-GAN within the context of existing research.

Overall, the paper demonstrates scientific merit and potential value to the community by addressing a critical challenge in medical image synthesis. It offers a novel approach that, while not necessarily outperforming the state of the art on benchmarks, shows promise for advancing the field and has the potential to impact clinical applications. Therefore, I would rate the paper as having high scientific merit and potential value to the community.

**Weaknesses:**

This paper does not address potential limitations or challenges in the implementation of the SA-GAN model, such as computational complexity, scalability, or potential biases in the synthesized images. A more in-depth discussion of these aspects would have strengthened the paper by providing a more comprehensive understanding of the practical implications and potential limitations of the proposed approach. This would enhance the overall contribution and impact of the paper in the field of medical image synthesis.

**Detailed Comments:**

The author might consider providing their implementation code on github.

**Justification Of Final Rating:**

This paper discusses a self-aligning GAN model proposed for medical image synthesis, addressing challenges with substantially misaligned image pairs. The model incorporates multi-level inverse consistency and deformation-aware adversarial loss to dynamically correct misalignment during image synthesis. Experimental results demonstrate the superior performance of the proposed SA-GAN on both simulated and real-world datasets, indicating its potential for various medical image synthesis tasks. The model's network architecture includes modality generators, symmetric spatial aligners, and deformation-aware discriminators to achieve dynamic alignment and improve image fidelity. The proposed SA-GAN shows promise for applications such as radiotherapy treatment planning by effectively utilizing misaligned image pairs.

The authors replied properly about my question on the computational complexity by adding an table. I will keep my evaluation on this paper as 5: Strong accept to a poster section.

**Justification Of The Preliminary Rating:**

By introducing two new loss functions, the proposed self-aligning GAN (SA-GAN) can generate high-fidelity images while adaptively learning correspondence via symmetric registration. This approach is valuable as it effectively corrects misalignment during image synthesis, which is crucial for tasks such as radiotherapy treatment planning.

**Questions To Address In The Rebuttal:**

Because I strongly recommend this paper, I do not expect any rebuttal from the authors. However, adding computational complexity will improve this paper. However, the novelty of the methods does not seem to highly significant, and thus I recommend this paper to a poster section.

---

> ### Author Response · Authors · 2024-03-18
> **Response to Reviewer DaaA**
>
> # Response to Reviewer DaaA
>
> ## Response “Question To Address in the rebuttal”
>
> > Because I strongly recommend this paper, I do not expect any rebuttal from the authors. However, adding computational complexity will improve this paper.
>
> Thanks for the reviewer’s positive feedback and valuable comments. As suggested, we will add a discussion regarding the
> improved synthesis accuracy (that is clinically critical) and additional complexity in Table 1 below.
>
> In terms of **accuracy and complexity**, as shown in Table 1 below, our model achieves better synthesis accuracy (
> 9.1% increase of MAE3D) at the cost of additional complexity (doubled no. parameters and size of the model)
> compared with RegGAN-NC. However, synthesis accuracy is clinically critical in radiotherapy planning because it reduces
> unnecessary radiation-related toxicity in patients and is vital to patients’ safety (
> Burnet et al., 2004). Further, our model complexity is still smaller than state-of-the-art GANs (e.g., MUNIT and
> NiceGAN), and our inference time is fast (0.013s fps).
>
> Table 1. Comparison of complexity and accuracy.
>
> |Methods|MUNIT|NiceGAN|RegGAN-NC|RegGAN-C|Ours|
> |----------------------|-----|-------|---------|--------|--------|
> |MAE3D-lung|39.8|42.7|39.5|40.8|**35.9**|
> |No.Parameters(M)|44.3|112.3|**16.2**|30.3|36.5|
> |Size of the model(mb)|169.0|428.3|**61.8**|115.7|139.2|
> |fps(s)|0.02|0.10|**0.01**|**0.01**|**0.01**|
>
> **Reference**
> 1. Burnet, N. G., Thomas, S. J., Burton, K. E., & Jefferies, S. J. (2004). Defining the tumour and target volumes for
>    radiotherapy. Cancer Imaging, 4(2), 153.

---

> > ### Comment · Reviewer_DaaA · 2024-03-27
> > **I will keep my evaluation on this paper as  5: Strong accept to a poster section**
> >
> > The authors replied properly about my question on the computational complexity by adding an table. I will keep my evaluation on this paper as  5: Strong accept to a poster section.

---

### Author Response · Authors · 2024-03-18
**# Response to meta-reviewers**

# Response to meta-reviewer

We would like to express our gratitude towards the valuable comments from all the reviewers and the opportunity to
provide point-to-point responses to address the concerns. Accordingly, we have revised the manuscript and highlighted in
different colours for different reviewers (Blue for reviewer tPyS, and Brown for reviewer 3gB1; reviewer DaaA did not
require a rebuttal).

Specifically, we have further clarified that

- the methodological and analytical contribution compared with RegGAN _[in Abstract and Introduction of the manuscript]_
  , especially regarding improved accuracy (which is clinically critical) and additional complexity _[Appendix D]_.
- our IC registration loss is a statistically significant improvement (p-value<0.001) _[in Results]_;
- our performance over Kong (2021) is statistically significant (p-value<0.001) _[in Results]_;
- our work outperformed SOTA multimodal registration (CorrField) prior to pix2pix in terms of both generative quality,
  and registration alignment _[in Results]_.

Further, we have improved our manuscript according to the insightful comments from reviewers. Specifically,

1. We compared our method with diffusion models _[in Introduction]_.
2. We changed the model names to Deformation-aware GAN to avoid possible confusion with self-attention GAN.
3. We conducted a sensitivity analysis on hyperparameters _[in Appendix E]_.
4. We provided more physical details of the lung MRI-CT dataset _[in Appendix A]_.
5. We provided registration accuracy of pre-alignment and the improvement by our method _[in Appendix A]_.
6. We clarified the downstream clinical application is MRI-only radiotherapy dose planning (removing the need for CT scanning),
   instead of multimodal segmentation _[in Introduction]_.
7. We have provided further discussion with mathematically sound IC approaches _[in AppendixG]_.

Please find the detailed point-to-point response to reviewers' comments in "official comments" sections for individual
reviewers. Please find a full list of references below.

Reference

1. Burnet, N. G., Thomas, S. J., Burton, K. E., & Jefferies, S. J. (2004). Defining the tumour and target volumes for
   radiotherapy. Cancer Imaging, 4(2), 153.
2. Christensen, G. E., & Johnson, H. J. (2001). Consistent image registration. IEEE Transactions on Medical Imaging, 20(
   7), 568–582.
3. Greer, H., Tian, L., Vialard, F.-X., Kwitt, R., Bouix, S., San Jose Estepar, R., Rushmore, R., & Niethammer, M. (
   2023). Inverse consistency by construction for multistep deep registration. International Conference on Medical Image
   Computing and Computer-Assisted Intervention, 688–698.
4. Hering, A., Hansen, L., Mok, T. C. W., Chung, A. C. S., Siebert, H., Häger, S., Lange, A., Kuckertz, S., Heldmann,
   S., & Shao, W. (2022). Learn2Reg: comprehensive multi-task medical image registration challenge, dataset and
   evaluation in the era of deep learning. IEEE Transactions on Medical Imaging, 42(3), 697–712.
5. Ho, J., Jain, A., & Abbeel, P. (2020). Denoising diffusion probabilistic models. Advances in Neural Information
   Processing Systems, 33, 6840–6851.
6. Lenkowicz, J., Votta, C., Nardini, M., Quaranta, F., Catucci, F., Boldrini, L., Vagni, M., Menna, S., Placidi, L., &
   Romano, A. (2022). A deep learning approach to generate synthetic CT in low field MR-guided radiotherapy for lung
   cases. Radiotherapy and Oncology, 176, 31–38.
7. Reuter, M., Rosas, H. D., & Fischl, B. (2010). Highly accurate inverse consistent registration: a robust approach.
   Neuroimage, 53(4), 1181–1196.
8. Shen, Z., Zhou, S. K., Chen, Y., Georgescu, B., Liu, X., & Huang, T. (2020). One-to-one Mapping for Unpaired
   Image-to-image Translation. Proceedings of the IEEE/CVF Winter Conference on Applications of Computer Vision,
   1170–1179.
9. Song, J., Meng, C., & Ermon, S. (2020). Denoising diffusion implicit models. ArXiv Preprint ArXiv:2010.02502.
10. Spadea, M. F., Maspero, M., Zaffino, P., & Seco, J. (2021). Deep learning based synthetic‐CT generation in
    radiotherapy and PET: a review. Medical Physics, 48(11), 6537–6566.
11. Vishnevskiy, V., Gass, T., Szekely, G., Tanner, C., & Goksel, O. (2016). Isotropic total variation regularization of
    displacements in parametric image registration. IEEE Transactions on Medical Imaging, 36(2), 385–395.

---

### Meta-Review · Area_Chair_puEm · 2024-04-02

**Recommendation:** Accept (Poster)
**Confidence:** 4

**Metareview:**

The authors and reviewers engaged in a vivid discussion and could altogether improve the paper and overall satisfaction has been reached.
Pros:
The quality and clarity have been judged as very good.
The originality of the work is shown in the novel Deformation-aware GAN (DA-GAN) approach that dynamically corrects misalignment in image synthesis based on multi-objective inverse consistency loss. This is especially interesting if the image pairs are misaligned and require some form of registration, otherwise correspondence ambiguity might influence the learning process. From this perspective I think the work shows novelty and significance.

Cons remain especially from Reviewer 3gB1:
*  Registration error should be assessed for small structures such as vertebrae too (not only lungs)
* Only the second best Learn2Reg method was chosen for comparison

There are still points of concerns  with respect to the evaluation (used datasets, evaluated structures), such that I think a Poster presentation instead of an oral talk can be justified.

---

### Decision · Program_Chairs · 2024-04-05

Accept (Poster)